# A New Optical Sensor Based on Laser Speckle and Chemometrics for Precision Agriculture: Application to Sunflower Plant-Breeding

**DOI:** 10.3390/s20164652

**Published:** 2020-08-18

**Authors:** Maxime Ryckewaert, Daphné Héran, Emma Faur, Pierre George, Bruno Grèzes-Besset, Frédéric Chazallet, Yannick Abautret, Myriam Zerrad, Claude Amra, Ryad Bendoula

**Affiliations:** 1ITAP, Univ Montpellier, INRAE, Institut Agro, 34000 Montpellier, France; daphne.heran@inrae.fr (D.H.); emma.faur@etu.umontpellier.fr (E.F.); ryad.bendoula@inrae.fr (R.B.); 2Innolea, 6 Chemin des Panedautes, 31700 Mondonville, France; pierre.george@innolea.fr (P.G.); bruno.grezes-besset@innolea.fr (B.G.-B.); 3Shakti, 45 Rue Frédéric Joliot-Curie, 13013 Marseille, France; shakti@shakti.fr; 4Aix Marseille Univ, CNRS, Centrale Marseille, Institut Fresnel, 13013 Marseille, France; yannick.abautret@fresnel.fr (Y.A.); myriam.zerrad@fresnel.fr (M.Z.); claude.amra@fresnel.fr (C.A.)

**Keywords:** optical sensor, precision agriculture, plant-breeding, laser speckle, chemometrics, multivariate data analysis, phenotyping, REP-ASCA

## Abstract

New instruments to characterize vegetation must meet cost constraints while providing accurate information. In this paper, we study the potential of a laser speckle system as a low-cost solution for non-destructive phenotyping. The objective is to assess an original approach combining laser speckle with chemometrics to describe scattering and absorption properties of sunflower leaves, related to their chemical composition or internal structure. A laser diode system at two wavelengths 660 nm and 785 nm combined with polarization has been set up to differentiate four sunflower genotypes. REP-ASCA was used as a method to analyze parameters extracted from speckle patterns by reducing sources of measurement error. First findings have shown that measurement errors are mostly due to unwilling residual specular reflections. Moreover, results outlined that the genotype significantly impacts measurements. The variables involved in genotype dissociation are mainly related to scattering properties within the leaf. Moreover, an example of genotype classification using REP-ASCA outcomes is given and classify genotypes with an average error of about 20%. These encouraging results indicate that a laser speckle system is a promising tool to compare sunflower genotypes. Furthermore, an autonomous low-cost sensor based on this approach could be used directly in the field.

## 1. Introduction

Crop phenotyping gives access to vegetation characteristics called phenotypic traits. New phenotypic traits other than yield-related or morphological traits have emerged. They can be directly or indirectly related to growth, development, architecture and to resistance, resilience and tolerance to biotic and abiotic stresses [1,2]. Quantifying a complex trait such as drought tolerance or adaptation to heat stress often requires a set of measurable indicators (secondary traits or indirect measurement) over the whole plant [2,3]. These indicators of interest must be measured accurately to be valuable for plant-breeding [4,5,6]. However, accessing this information directly to the field is cost- and time-consuming [7]. For this reason, phenotyping for plant-breeding purpose is limited.

Phenotyping using an optical instrument is a mean to provide phenotypic information in an objective, rapid and non-destructive manner [8,9]. Presently, many optical techniques are an integral part of phenotyping procedures such as multispectral imaging and NIR spectrometry [10,11]. The most common approach is to propose indices based on a combination of a few spectral bands and related to biochemical variables, biomass or photosynthetic activity. Associated with a vector, pedestrian, tractor or drone, these technologies demonstrate the feasibility of relatively inexpensive tools for high-throughput phenotyping [8,9]. As phenotypic traits have diversified, innovative low-cost optical instruments need to be developed to better describe new genotypes.

In recent years, the use of laser speckle has developed mainly in the biomedical field [12,13] and very recently in the agricultural field [14,15]. Laser speckle imaging is a low-cost, sensitive and noninvasive method. A typical setup for speckle measurement is very simple. It requires an expanded laser light, which might be a diode laser, a detector such as a CCD camera with a lens and a PC with a frame grabber that is able to record a set of images. These features make this method attractive to many applications, which require fast and non-destructive sampling. Combined with polarization, speckle image analysis techniques provide new parameters describing media properties (such as scattering or absorption) [16,17,18,19,20,21] which makes this method very attractive for precision agriculture applications.

Leaf scattering and absorption parameters are related to its chemical composition and structure (cuticle, epidermis, mesophyll) [22] which is the seat of the photosynthesis process [23]. They are likely to vary according to the onset of stress or disease, resulting in leaf dryness or wilting for example.

As laser speckle is sensitive to scattering and absorption properties, its measurement is likely to identify structural variations between plants. Statistical parameters are computed from speckle images and provide information on the surveyed sample. These statistical parameters are then potential indicators usable for non-destructive plant phenotyping.

All the statistical parameters constitute a so-called multivariate observation. To analyze a set of speckle measurements associated with a design of experiments, methods of analysis of variance must be adapted to multivariate data. The method called Reduction of Repeatability Error-Analysis of Variance-Simultaneous Component Analysis (REP-ASCA) [24] is suitable to study the influence of identified factors while reducing error due to the lack of repeatability of measurements.

In this paper, we propose an original method to assess the potential of laser speckle measurements analyzed with chemometrics as a non-destructive phenotyping tool. In particular, sensitivity to physiological criteria, such as inner structure, will be discussed. An optical system combining laser speckle and polarization at two wavelengths will be tested on a case study: the discrimination of sunflower genotypes under identical water conditions.

The objectives are (1) to characterize possible sources of error degrading the speckle measurements, (2) identify variables involved in genotype dissociation, (3) to establish a genotype discrimination model based on these variables.

## 2. Materials and Methods

### 2.1. Data Acquisition and Experiment

#### 2.1.1. Speckle Measurement

The optical setup used for backscattered speckle measurements is schematically illustrated in Figure 1. Two laser diodes were used in order to stimulate the samples at two wavelengths: one at 660nm (Thorlabs HL6545MG) where both absorption and scattering occurred and one at 785nm (Thorlabs L785P100) where absorption was negligible [25,26]. Both laser diodes were mounted with an aspheric lens (Thorlabs C220TME-B for the 660nm laser diode and Thorlabs C330TME-B for the 785nm laser diode) adjusted to set the laser spot to 5 mm-diameter at the sample surface and optical densities to set the power to 8mW at 660nm and 15mW at 785nm.

A grid polarizer (Thorlabs WP12L-UB) was mounted between the laser diode and the sample to set a p-polarization and an analyzer (Thorlabs WP25M-UB) mounted in front of the camera (CMOS, Thorlabs DCC3240M) to measure backscattered speckle in p- and s-polarization alternatively. The CMOS camera recorded the speckle field on 1024×1280 pixels of 5.3μm × 5.3μm pitch. The distance between the sample and the camera was set to 20cm so a typical speckle spot covered a few pixels. The image measured with parallel polarizer and analyzer was named *–pp* and the image measured with crossed polarizer and analyzer was named *–ps*.

The integration time of the camera was set to 3ms at 660nm and 1ms at 785nm and the frame rate was set to 54 fps to avoid blur on the image due to particle motion in the biological sample. For each measurement, 30 frames were taken in a row. A dark measurement was also performed for each sample to remove the background signal on the images.

#### 2.1.2. Experimental Design

A design of experiments was built to compare four sunflower genotypes in comfortable water and light conditions. Sunflower plants were grown in a greenhouse at INRAE, France. Water and lighting conditions were similar for each pot with a day-night cycle of 12 h/8 h. The greenhouse was equipped with multispectral lighting (450nm, 560nm, 660nm, 730nm and 6000${}^{\circ }$K) controlled by Herbro automaton (GreenHouseKeeper). Irrigation occurred every 2 days and corresponded to water comfort condition.

For the four selected genotypes, two potted plants of each were grown. Four leaves were collected at the upper and middle parts of each plant. On each leaf, six regions of interest (ROIs) were selected and measured with our setup. As a result, 4×2×4×6=192 images were acquired. Therefore, the identified factors of this experimental design are genotype, leaf and zone. This dataset was divided into two sets, one part to form an independent test set and a second part to form a calibration set. The independent test set was defined by the observations of ROIs on two leaves from each of the four genotypes. As a result, the independent test set was composed by 4×2×6=48 observations. The calibration set was then constituted with the remaining observations, i.e., 144 observations.

### 2.2. Data Analysis

#### 2.2.1. Polarized Speckle Parameters

Several statistical parameters related to the speckle pattern for each polarization can be extracted from the speckle images. These parameters were computed for the 30 frames and then averaged on the number of realizations to give the results for one measurement. First of all, the average intensities of the images −*pp* and −*ps* were computed: $<I_{pp}>$, $<I_{ps}>$. $<I_{pp}>$ is the sum of the polarization maintaining light and half of the depolarized light, whereas $<I_{ps}>$ corresponds to half of the depolarized light [27]. The average surface and volume intensities, respectively $I_{surf}$ and $I_{vol}$ were then defined as:(1)$$I_{surf}=<I_{pp}>-<I_{ps}>$$
(2)$$I_{vol}=2\;.<I_{ps}>$$

The degree of linear polarization ($DOP_{l}$), which describes the portion of the electromagnetic wave which is polarized, can also be computed to characterize samples. It was given by:(3)$$DOP_{l}=\frac{<I_{pp}>-<I_{ps}>}{<I_{pp}>+<I_{ps}>}$$

Moreover, in [28,29] the “average width” of a speckle pattern is determined from calculations of the normalized autocorrelation function of the intensity distribution in the (x,y) plane. This function, denoted $c_{I}\left(\Delta x,\Delta y\right)$ was calculated from the intensity distribution of the measured speckle, *I(x,y)*:(4)$$c_{I}\left(\Delta x,\Delta y\right)=\frac{FT^{-1}\left[{\left|FT\left[I\left(x,y\right)\right]\right|}^{2}\right]-<I\left(x,y\right){>}^{2}}{<I{\left(x,y\right)}^{2}>-<I\left(x,y\right){>}^{2}}$$
where *FT* is the Fourier Transform, < . > is a spatial average. We define $c_{I}\left(\Delta x,0\right)$ and $c_{I}\left(0,\Delta y\right)$ the horizontal and vertical profiles of $c_{I}\left(\Delta x,\Delta y\right)$, respectively. The full width at half maximum of this function provides a reasonable measure of the speckle size. Given the setup geometry, only the width along the vertical profile (dy such as $c_{I}\left(0,dy/2\right)=0.5$) was considered, to avoid the incident angle influence. For each sample, two speckle sizes were computed in micrometers, depending on the analyzed polarization state: $dy_{pp}$ and $dy_{ps}$. Moreover, the difference $dy_{pp}$ − $dy_{ps}$ has been investigated in the case of volume scattering and absorbing media characterization [20].

One last parameter extracted from speckle images *–pp* or *–ps* was the contrast ($C_{pp}$ or $C_{ps}$) defined as [28]:(5)$$C=\frac{\sigma_{I}}{<I(x,y)>}$$
where $\sigma_{I}$ is the square root of the intensity variance and < . > is a spatial average.

#### 2.2.2. Multivariate Data Analysis

In this study, an observation is defined by a number *p* of parameters (here *p* = 20) extracted from a speckle image. The observations of an experiment can be represented by a matrix X of dimension n×p, where *n* is the number of observations.

##### REP-ASCA

An analysis of variance method called Reduction of Repeatability Error-Analysis of Variance-Simultaneous Component Analysis (REP-ASCA) [24] was used (1) to assess whether the genotype has a significant impact on speckle measurement, (2) to identify variables involved in this differentiation.

This method, derived from the Analysis of Variance-Simultaneous Component Analysis (ASCA) method [30] can take into account the lack of repeatability of measurements. Indeed, when factors are not identified or nested in the experimental design, measurements are likely to vary. These variations can then alter conclusions of an analysis of variance.

REP-ASCA approach consists of distinguishing the multivariate dataset X dedicated to the analysis of variance from a matrix W describing the dataset repeatability error. W dimension is m×p, *m* being the number of observations for this dataset (separate from the *n* observations constituting X). To this end, the matrix X is projected orthogonally to the *k*-first principal components of W, resulting in a matrix called $X_{⊥}$:(6)$$X_{⊥}=X\left(I-DD^{t}\right)$$
where I is the identity matrix of dimension *p* and D is a p×k matrix containing the *k*-first principal components of W.

ASCA is then performed on $X_{⊥}$ which no longer contains the information carried by the *k*-first principal components of W. Moreover, $X_{⊥}$ can be decomposed into a sum of observation matrices related to identified factors. For example, with a number *i* of identified factors, the decomposition is written as follows:(7)$$X_{⊥}=\mu +\sum_{i}X_{i}+R$$
where μ is the average matrix of $X_{⊥}$, $X_{i}$ the observation matrix corresponding to the factor *i* and R the residuals.

Additionally, permutation tests are performed to evaluate whether identified factors are significant or not on the dataset overall variance [31]. Variances are obtained by calculating the sum of squares (SSQ). For $X_{⊥}$, its variance is defined as follows:(8)$$\mathrm{SSQ}\left(X_{i}\right)={∥X_{i}∥}^{2}$$
where ${∥.∥}^{2}$ indicates Frobenius’ norm, i.e., the sum of squares of the matrix elements. Finally, REP-ASCA provides the loadings of each factor principal components and the corresponding observation scores. In the example given in Equation (Equation 7), considering *l* principal components, $X_{i}$ is then decomposed into a matrix $P_{i}$ (of dimension p×l) containing the loadings and a matrix $T_{i}$ (of dimension n×l) containing the scores. With the matrix $R_{i}$ containing the residuals, the equation is written:(9)$$X_{i}=T_{i}P_{i}^{t}+R_{i}$$

In our study, REP-ASCA method was tested on the calibration set (144 observations) where 96 observations (=*n*) were used to form X. The remaining 48 observations (=*m*) were used to form W. Thereafter, X was projected orthogonally to the *k*-first principal components retained from W to give $X_{⊥}$ (Equation (Equation 6)). $X_{⊥}$ was decomposed according to the model established as follows:(10)$$X_{⊥}=\mu +X_{G}+X_{L}+X_{Z}+X_{G\times L}+X_{G\times Z}+X_{L\times Z}+X_{G\times L\times Z}+R$$
where μ is the average matrix of $X_{⊥}$. The terms $X_{G}$, $X_{L}$ and $X_{Z}$ are matrices corresponding respectively to the genotype, leaf and zone factors. The terms $X_{G\times L}$, $X_{G\times Z}$, $X_{L\times Z}$ and $X_{G\times L\times Z}$ are respectively the interaction terms of genotype/leaf, genotype/zone, leaf/zone and genotype/leaf/zone. The R matrix represents the residuals.

## 3. Results and Discussion

### 3.1. Speckle Images

For each ROI on a leaf, speckle patterns are acquired at 660nm and 785nm, for *pp*- and *ps*-polarizations. Figure 2a,b show examples of these patterns at 660nm. The average intensity of the image is higher for *pp*-polarization (Figure 2a) than for *ps*-polarization (Figure 2b). Indeed, the absorption of electromagnetic radiation at 660 nm is very high (absorption of chlorophyll [32]). The longer the path in the surveyed medium, the greater the absorption. Image acquired in *ps*-polarization corresponds to light having traveled inside the sample volume [21,27], explaining why the average intensity is lower.

At 785nm, speckle images for *pp*-polarization (Figure 3a) and *ps*-polarization (Figure 3b) have similar value scales. At this wavelength, leaves absorption is close to zero. Light is therefore mainly scattered and lose its polarization. The proportion of light in parallel and cross polarizations is then similar.

### 3.2. Multivariate Data Analysis

#### 3.2.1. Repeatability Error Reduction

##### Selection of the *k*-First Components of the Repeatability Error

Principal components related to repeatability error are obtained from the matrix W defined in Section 2.2.2. The influence of the number *k* of retained components on the analysis of variance results of $X_{⊥}$ is studied. Total variance (Figure 4a) and percentages of explained variance by factor (Figure 4b) of the dataset $X_{⊥}$ are the two selected criteria to choose the value of *k*.

REP-ASCA reduces the error due to lack of repeatability, resulting in a decrease in the percentage of explained variance of residuals, and an increase in the variance explained by factors. However, the orthogonal projection also removes some of the information carried by the principal components. It is, therefore, necessary to select a value of *k* as small as possible.

When *k* = 0, no projection has been made and the results correspond to those of the ASCA method. The percentage variance of the residuals has a value of 55.5% and that of the genotype term is 15.4% (Figure 4b). The proportion of residuals is important: this may come from unidentified factors such as phenomenological stage or variations in measurement conditions.

In addition, the genotype factor seems to induce greater changes in speckle parameters than the leaf or zone factor and all interactions between them (Figure 4b). The percentage of explained variance of the genotype term increases when the first three components are removed to reach a value of 22.0% and then decreases when the fourth component is removed. We therefore choose to remove the first three components to minimize the percentage of variance explained by the residuals and maximize the percentage of variance explained by the genotype term while avoiding to strongly reduce the total variance.

##### Description of the *k*-First Components of the Repeatability Error

The loadings of the first three components of the matrix W describing the repeatability error are visible on correlation circles in Figure 5. These loadings are presented in the plane formed by the first two principal components (Figure 5a) and in the plane formed by the first and third components (Figure 5b).

On the first component (Figure 5a,b), the loadings of the variables obtained at 785nm are all negative with high values for $DOP_{l}$, $I_{surf}$ and $<I_{pp}>$. Whatever the wavelength, $I_{surf}$ and $<I_{pp}>$ parameters may correspond to unwilling specular reflections due to the fact that the leaf surface is not perfectly flat. The $DOP_{l}$ characterizes the portion of light that has kept its polarization state. Specular reflections can increase this parameter as well. The orthogonalization of the dataset X to this first component of error would thus reduce the part of specular reflections at 785nm which carries little information on the differences between the four genotypes.

On the second component (Figure 5a), the loadings are high for the variables measured at 660nm. In particular, the variables $dy_{pp}-dy_{ps}$ and $dy_{pp}$ have high negative values while $<I_{pp}>$, $DOP_{l}$ and $I_{surf}$ have high positive values. This second component of W therefore reflects large variations on the variables measured at 660nm. These variations may be related, on the one hand, to the chemical composition of the leaf, such as chlorophyll content, the variables $dy_{pp}-dy_{ps}$ and $dy_{pp}$ being related to absorption [21]; and, on the other hand, to undesirable specular reflections through the variables $<I_{pp}>$, $DOP_{l}$ and $I_{surf}$. The orthogonalization of X with respect to this second component reduces parts of variance associated with two sources of error. First, the absorption at 660nm, linked to the chlorophyll content, which is not a discriminating element between the genotypes of our study. Secondly, the specular reflections at the 660nm, which again, do not allow genotypes to be differentiated.

The loadings of the third component are visible in Figure 5b. As for the second component, the highest loadings are obtained at 660nm: $<I_{ps}>$, $C_{pp}$ and $I_{vol}$ in positive values and $C_{ps}$ in negative. At 660nm, the signal measured from the sample volume is weak because of the strong absorption at this wavelength. The variables $<I_{ps}>$, $I_{vol}$ or $C_{ps}$ are derived from the signal having traveled deep into the leaf [21], they will therefore be impacted by the strong absorption and will present a low signal-to-noise ratio. This explains why they are considered to be sources of error in the analysis of variance. The orthogonalization of X with respect to this third error component reduces the weight of these imprecise parameters.

The lack of repeatability of measurements is reflected in the information carried by the loadings of the first three principal components described above. The study of these components showed that the repeatability error was described by parameters related to undesirable specular reflections at 785nm and 660nm, as well as variations induced by the strong absorption of the leaf at 660nm.

#### 3.2.2. Dataset Orthogonalization and Description of the Genotype Factor

Components related to the repeatability error described in the previous section are used to orthogonalize X to study the factors described by $X_{⊥}$. To ensure the significance of these factors, a permutation test is first performed. In addition, finally, in the last step of REP-ASCA, these factors are decomposed into loadings and scores. Thus, for the genotype factor, loadings of the principal components identify variables involved in inter-genotype variability and scores can separate or gather genotypes.

##### Permutation Test

For each factor of $X_{⊥}$, a permutation test is performed with 2500 random permutations of level assignments (Figure 6).

Null hypothesis distributions formed by the assignment permutations are represented by histograms while the variance assigned to the studied factor is identified by the red dot. The genotype factor (Figure 6a) and the leaf factor (Figure 6b) are significant. However, the zone factor and the genotype/leaf interaction factor (Figure 6c,d) are not. This means that whatever the leaf, the genotype has a strong significant impact on the variables obtained by speckle image. We are then likely to be able to differentiate genotypes whatever the leaf chosen.

##### Loadings of the Genotype Factor

The genotype factor has four levels corresponding to the four studied genotypes. The principal component analysis of this factor then provides three components whose loadings are visible Figure 7. The percentages of variances explained by these components are 60.6%, 30.9% and 8.5% for the first, second and third principal components, respectively. Correlation circles show the loadings on the first and second principal components (Figure 7a) and on the first and third principal components (Figure 7b).

On the first component (PC1), strong negative values are visible for the variables measured at 785nm: $<I_{ps}>$, $I_{vol}$, $C_{ps}$ and $<I_{pp}>$. Moreover, the variable $dy_{ps}$ at 660nm has the highest positive value. According to [21], the variable $dy_{ps}$ is strongly related to scattering, even in the presence of absorption. Moreover, PC1 highlights variables measured at 785nm and in particular those related to the interaction of light in the sample volume (*ps*-polarization or volume intensity). At this wavelength, scattering is much more important than absorption in the leaf [33]. These variations in intensity between genotypes are therefore related to light scattering properties within leaves. For example, internal structure or changes in optical indices within leaf are elements likely to modify its scattering properties [34].

On the second component (PC2), the most important values are obtained for 3 variables measured at 785nm: $C_{ps}$ and $dy_{pp}-dy_{ps}$ with positive values and $dy_{ps}$ with a strong negative value. Again, PC2 seems to highlight scattering properties within the leaves. The variable $dy_{ps}$ at 785nm is related to scattering properties [21]. The same is true for the variable $dy_{pp}-dy_{ps}$ at 785nm, which reflects scattering differences between the surface (*pp*-polarization) and the volume (*ps*-polarization) of the sample. We can notice that on PC1, the variables selected were rather related to intensity levels backscattered by the leaf at 785nm, while PC2 highlights variables related to geometrical properties of speckle spots (*dy* being inversely proportional to the spread of the light spot on the sample). In both cases, these variables are related to the scattering properties of the sample and thus to leaf structure.

The third component (PC3) of the genotype factor carries little variance compared to the first two (8.5%). The loadings of variables measured at 660nm are comparable to those obtained on PC1, in particular with a high value for the variable $dy_{ps}$. However, unlike for PC1 and PC2, no remarkable variable can be identified.

##### Scores for Each Genotype

Scores are obtained from the projection of the observation set on the three components of the genotype factor. For each of the four studied genotypes, the average score and its confidence ellips [35], representing the variability within a same level, are presented in Figure 8a,b.

Genotypes A and B are separated from genotypes C and D on PC1 (Figure 8a). As mentioned above (Figure 7a), PC1 is related to bulk variables at 785nm with negative loadings for $<I_{ps}>$ or $I_{vol}$. Consequently, negative scores on this component (Figure 8a) reflect high values for these variables, highlighting a strong scattering from the sample volume. This is what is observed for genotypes A and B whereas genotypes C and D show positive average scores on PC1. The discrimination on PC1 between these two groups of two genotypes could therefore be based on the difference in leaf internal structure.

Furthermore, PC2 separates genotype C from genotype D (Figure 8a,b). This discrimination is mainly due to $dy_{ps}$ at 785nm (high negative loading, see Figure 7a), which represents the typical size of speckle spots. This variable is strongly correlated with scattering properties [21] and thus to leaf physical structure. In addition, the confidence ellipses of genotypes C and D do not overlap. It is then possible to discriminate these two genotypes using PC2.

Finally, PC3 (Figure 8b) can be used to distinguish genotype A from genotype B. However, the confidence ellipses of these two genotypes overlap. The discrimination of these two genotypes is therefore not faultless.

In this section, the genotype factor of $X_{⊥}$ has been decomposed into three principal components, to study speckle variables most likely to distinguish the four genotypes. The first two components PC1 and PC2 account for a total of 91.5% of the dataset variance. The speckle variables highlighted by these components are mainly those reflecting variations in light scattering within the leaf at 785nm.

After projection on PC1, PC2 and PC3 of all observations, the average scores obtained show the feasibility of discriminating the four sunflower genotypes. This discrimination is based on the leaf structural properties impacting the measured speckle variables.

#### 3.2.3. Example of Application to Discriminant Analysis

Reducing a dataset dimensionality is a common step prior to discriminant analysis (DA) [36,37]. In this example, a DA is performed into the subspace formed by the components of the genotype term (Figure 7), i.e., calibration set and test set are both projected onto these components provided scores. The DA model was calibrated using scores obtained from the calibration set and applied on scores from the test set. The purpose of this example is to check the ability of the principal components formed by the speckle parameters to discriminate genotypes.

The proposed method is applied to an independent test set composed of 48 observations (defined in Section 2.1.2). The three principal components of $X_{G}$ are used in a discriminant analysis to classify genotypes. First, the discrimination model is calibrated with the scores obtained previously on the calibration set (Figure 8a,b). Then, the 48 observations of the test set are projected on these three principal components to give the corresponding test scores. Finally, the discrimination model is applied on these test scores and yields the confusion matrix shown in Table 1.

The average error of discrimination is approximately 20%. Genotype D has a perfect ranking (12/12). Genotypes A (9/12) and B (10/12) are also well ranked. Genotype C has the lowest ranking (7/12). These results are in good agreement with the analyses of the average scores (Figure 8a,b).

The results of this discriminant analysis demonstrate the potential of our polarization and speckle setup for genotype discrimination. The average error of discrimination of about 20% is encouraging for this first experimental campaign.

## 4. Conclusions

In this paper, we investigated the potential of laser speckle measurements as a plant phenotyping tool. We were interested in the discrimination of sunflower genotypes.

The combination of laser speckle and polarization at two wavelengths has provided us a set of 20 parameters that can be used as multivariate observations. Using the analysis of variance method called REP-ASCA, we demonstrated that the genotype factor was strictly significant. Therefore, these parameters can be used to differentiate genotypes. We also identified the sources of error due to the lack of repeatability. Mostly, the analysis showed that the errors come from unwilling specular reflections on the leaf and from the variation of leaf chlorophyll content.

With the same method, studying the loadings of the genotype factor principal components pointed out the importance of variables measured at 785nm. These variables are mainly related to scattering properties within the leaf. Based on scores on these principal components, a DA can discriminate genotypes with an average error of 20% showing the ability of this new setup to distinguish genotypes.

Speckle measurements were performed at two selected wavelengths where different types of interactions with vegetation leaves occurred. Other wavelengths, specific to water, anthocyanins or carotenoids for example, could be added to improve the discrimination ability of the proposed approach. Moreover, for different genotypes, scattering and absorption properties are expected to vary heterogeneously in the presence of stress. We can make the assumption that the speckle pattern measured on leaves with the proposed setup will vary as well. Therefore, the next step will be to address the feasibility of using laser speckle to compare genotype responses to biotic or abiotic stresses.

## Figures and Tables

**Figure 1 sensors-20-04652-f001:**
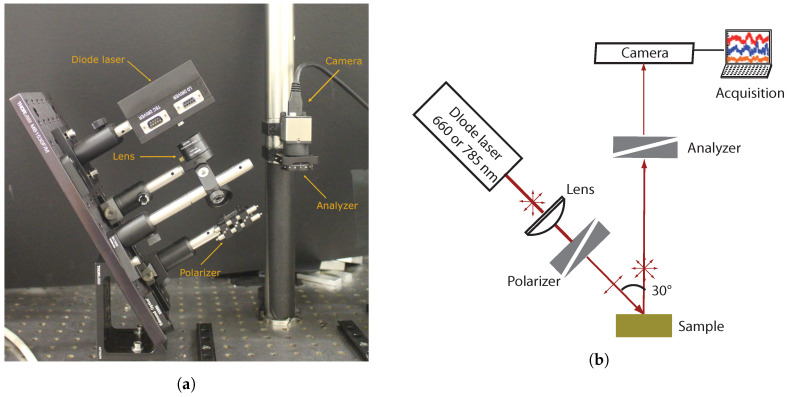
Experimental setup of speckle measurements: (**a**) image and (**b**) scheme.

**Figure 2 sensors-20-04652-f002:**
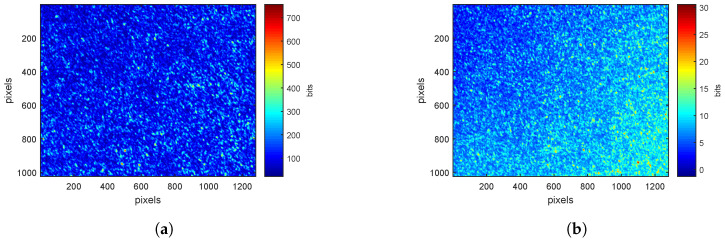
Speckle patterns at 660nm for (**a**) *pp*-polarization (**b**) *ps*-polarization with a color scale defined by the minimum and maximum values of pixels.

**Figure 3 sensors-20-04652-f003:**
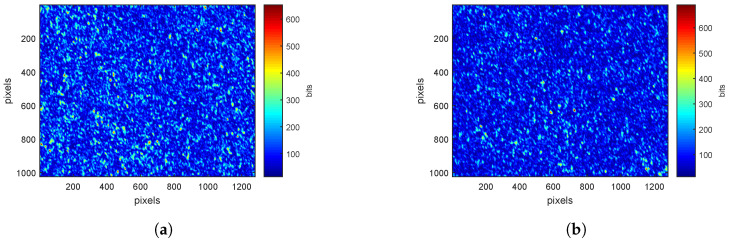
Speckle patterns at 785nm for (**a**) *pp*-polarization (**b**) *ps*-polarization with a color scale defined by the minimum and maximum values of pixels.

**Figure 4 sensors-20-04652-f004:**
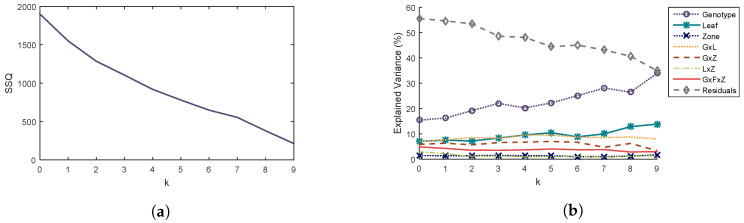
Influence of the number *k* of components taken into account for the orthogonal projection on (**a**) total variance SSQ of $X_{⊥}$ and (**b**) percentages of variance explained for each factor.

**Figure 5 sensors-20-04652-f005:**
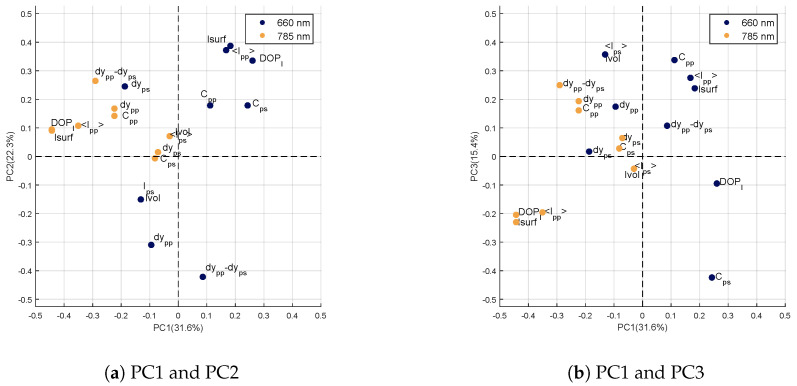
Correlation circle of the first three components of W.

**Figure 6 sensors-20-04652-f006:**
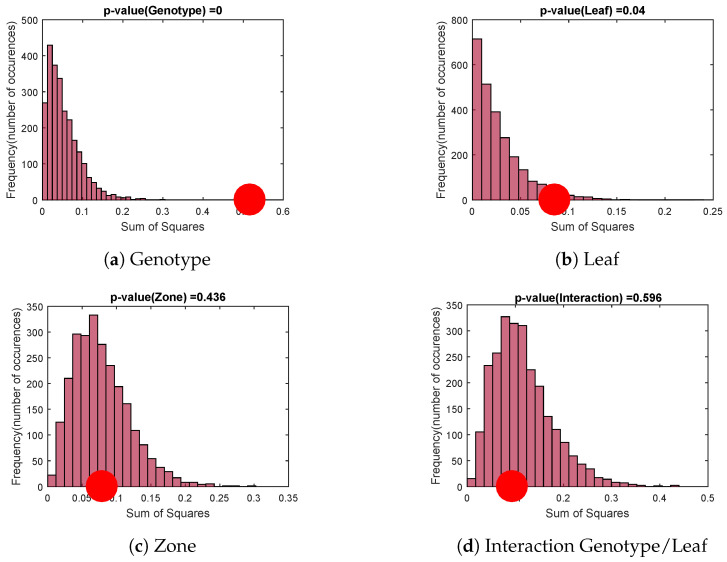
Permutation tests: comparison of the factor variance (red dot) with variances obtained by random permutations of level assignments.

**Figure 7 sensors-20-04652-f007:**
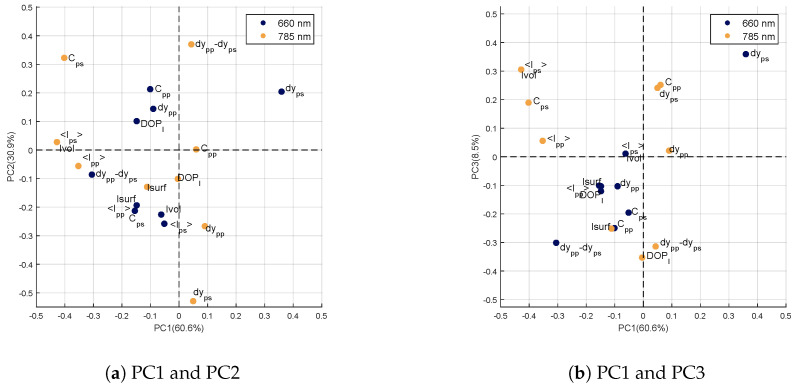
Correlation circle of the first three principal components of the term $X_{G}$.

**Figure 8 sensors-20-04652-f008:**
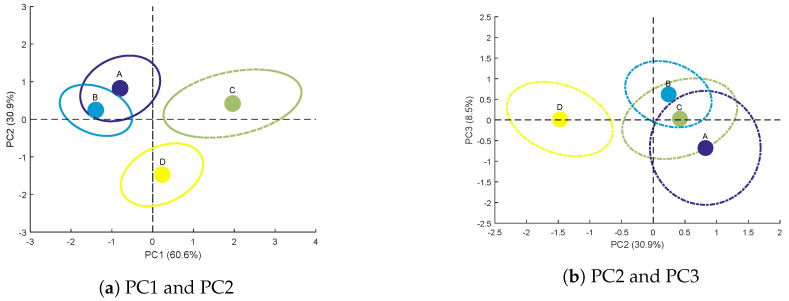
Average scores and confidence ellipses of the calibration set for each genotype on the principal components of the genotype factor.

**Table 1 sensors-20-04652-t001:** Confusion matrix of genotypes A, B, C & D.

	A	B	C	D
A	9	1	2	0
B	1	10	0	0
C	2	1	7	0
D	0	0	3	12

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
