# Peer review of "A New Optical Sensor Based on Laser Speckle and Chemometrics for Precision Agriculture: Application to Sunflower Plant-Breeding"

_sensors, 2020, doi:10.3390/s20164652_

Round 1
Reviewer 1 Report
In this manuscript, the authors proposed a study of a simple two-wavelength laser speckle system dedicated for phenotyping. Scattering and absorption properties of sunflower leaves are related to 4 genotypes.
Authors should consider correction of units at bar scale on Figure 2 and 3. In my understanding, the number of bits is rather related to bit depth -parameter determined by the number of bits used to define number of colours in each pixel. Your camera is monochrome, if its working in 10 bits then each pixel may deliver 1024 grey levels.
The manuscript presents complete study at high level. I suggest to accept article in present form.
Kind regards,
Reviewer
Reviewer 2 Report
This manuscript reports on laser speckle for assessment of breeding process via the processing of two-wavelengths-captured speckle patterns. This is another application of this biospeckle technique in crops and leafs.
The manuscript is well-written, some minor typos and English errors are found over the text.
While the manuscript is interesting, still the sensors-related content is not the focus nor the novelty highlight. The core of this research is mainly the processing method. Little aspects of the technical setup justify that this publication is relevant for Sensors. I will let the Editor decide on the suitability of this manuscript for this journal, I would rather like to read such a research in a signal processing centered journal.
The authors must stress what makes their method any better with respect to what is present in literature and argument how their approach outperforms the others’. A fair comparison is missing in the manuscript. Otherwise, this results will be a so what? Paper. I think more arguments need to be provided on this front.
Some questions on the approach:
- A reference is missing in the phrase ending by “…NIR spectroscopy” at the end of page 1.
- What is the profile of the beams projected onto the sample? How did the authors ensure that the intensity and irradiances are not varying after expanding the beams?
- The authors state “Two laser diodes were used in order to stimulate the samples in two different wavelength ranges: one at 660 nm (Thorlabs HL6545MG) where both absorption and scattering occurred and one at 785 nm (Thorlabs L785P100) where absorption was negligible” but a reference is missing here to justify no absorption at 785 nm.
- The setup description is not as informative as it could be. Can the authors represent how the sample is placed for the monitoring? I would like to see a real image of the setup in the real measurement being performed.
- The first phrase on Page 4 is mixing electromagnetism with quantum physics by referring to photons, whereas the speckles are issued from optical physics. Please correct this.
- I would prefer to convert Figure 3 images to jet instead of presenting them in parula style. Also, consider normalizing the images or explaining the intensity variations due to the polarization shift.
- Figures 5, 7 and 8 contains the French term for and (et) in both a) and b). Correct this.
- Units are missing in Figure 6 a-d y-axis. Integrate them in the label.
English corrections:
Page 3 line 63 à correct “to stimulate the samples with two wavelengths” instead of “to stimulate the samples in two wavelength ranges”. Monochromatic sources normally are referred to using linewidth and not range or band.
Round 2
Reviewer 2 Report
The authors revised their original submission and this v2 version looks much better than the former one.
Some remarks were not addressed and I regret that the replies provided are not sufficient to accept this v2 in its current form.
- After reading the –new- Introduction I still do not see what encouraged the authors to adopt laser speckles are a new potential method to be implemented in breeding assessment. So, in my view a comparison is still missing even if the argument of this being a preliminary is presented. Why not just using one of the established –reference- methods instead? This is the argument I want to see in the manuscript’s Introduction and requires some comparison, which I think, by the way, is simple to justify.
- The real images in the reply letter are to be incorporated in Fig. 1 together with the scheme. I regret that the authors did not describe the components in between (#s) in the real images. I would suggest to take a new photo of the setup without the components not belonging to the laser speckle setup which are in the back of the image but still on the optical table.
- The phrase “to stimulate the samples in two wavelengths” in red on page 3 must replace in by at.
- The phrase “who describes the portion of the electromagnetic wave which is polarized” in red on page 4 must replace who by which.
Author Response
Please see the attachment,
Best regards,
